# The Safety and Efficacy of Calcitonin Gene-Related Peptide (CGRP) Monoclonal Antibodies for the Preventive Treatment of Migraine: A Protocol for Multiple-Treatment Systematic Review and Meta-Analysis

**DOI:** 10.3390/ijerph19031753

**Published:** 2022-02-03

**Authors:** Jaime Fernández-Bravo-Rodrigo, Carlos Pascual-Morena, Amparo Flor-García, Alicia Saz-Lara, Irene Sequí-Dominguez, Celia Álvarez-Bueno, Dolores Barreda-Hernández, Iván Cavero-Redondo

**Affiliations:** 1Health and Social Research Center, Universidad de Castilla—La Mancha, 16071 Cuenca, Spain; jaime.fernandezbravo@alu.uclm.es (J.F.-B.-R.); alicia.saz1@alu.uclm.es (A.S.-L.); irene.sequidominguez@uclm.es (I.S.-D.); celia.alvarezbueno@uclm.es (C.Á.-B.); ivan.cavero@uclm.es (I.C.-R.); 2Pharmacy Service, Hospital Virgen de la Luz, 16002 Cuenca, Spain; amparof@sescam.jccm.es (A.F.-G.); dbarreda@sescam.org (D.B.-H.); 3Universidad Politécnica y Artística del Paraguay, Asuncion 001518, Paraguay

**Keywords:** monoclonal antibodies, calcitonin gene-related peptide (CGRP), migraine prevention, erenumab, eptinezumab, fremanezumab, galcanezumab, protocol, review, meta-analysis

## Abstract

Background: Migraine is a common and disabling primary headache disorder, associated with many medical comorbidities, highly prevalent, with complex treatment and management. Currently, monoclonal antibodies targeting the trigeminal sensory neuropeptide, calcitonin gene-related peptide (CGRP), are available. The aim of this protocol is to provide a review comparing the effects and safety profile of different monoclonal antibodies in migraine patients. Methods: The literature search will be performed through the MEDLINE, Embase, CENTRAL, ClinicalTrials.gov, the WHO International Clinical Trials Registry Platform (ICTRP), Web of Science and Scopus databases, following the PICO strategy. Real World studies and randomized clinical trials assessing the effect of monoclonal antibodies against CGRP interventions (erenumab, eptinezumab, fremanezumab and galcanezumab) on monthly migraine days (MMD), monthly headache days (MHD), headache impact test (HIT-6) and triptan days of use (TriD) will be included. In Real World studies, the DerSimonian and Laird method will be used to calculate pooled estimates of the mean change difference and in randomized clinical trials, a network meta-analysis will be performed to estimate the comparative effects of different monoclonal antibodies against CGRP. Results: The findings of this study will be reported in a peer-reviewed journal. Conclusions: This study will provide evidence to health professionals on the efficacy and safety of different monoclonal antibodies against CGRP on the outcomes studied.

## 1. Introduction

Migraine is a common and often debilitating neurological disorder that is accompanied by a disabling primary headache with a plethora of transient somatosensory and motor disturbances [1,2,3]. Migraine diagnostic criteria distinguish between episodic and chronic migraine, with or without aura [1]. According to clinical experiences and patient reports, an acute migraine attack is triggered by a wide range of genetic (there are well-described genetic variants of migraine, such as FHM as monogenic migraine, and in contrast to FHM, the most common forms of migraine are result of the combined effects of multiple genes; polygenic migraine) and environmental factors (stress being the most frequent in 80% of cases) [4,5,6,7]. It is highly prevalent, affecting approximately 12% of the general population, 18% of women and 6% of males each year [8]. Migraine headaches are a leading cause of disability, suffering and economic burden to societies worldwide, ranking as the sixth leading cause of lost years in 2013 [9]. Migraine treatment and management is complex. Moreover, migraine patients with debilitating pain are 16 times more likely to overuse analgesics [10]. In moderate-severe acute migraine, triptans are the treatment of choice, but they are expensive and, although generally well tolerated, involve overuse and several contraindications [11,12]. There are wide range of preventive treatment options for reducing migraines, including beta blockers, antiepileptics, calcium channel blockers, antihypertensives, onabotulinum toxin-A, none of them with a specific mechanism of action in migraine [13].

However, the treatment of migraines is on the cusp of a new era with the development of drugs targeting the trigeminal sensory neuropeptide calcitonin gene-related peptide (CGRP) or its receptor, which has been shown to play a role in the onset of migraine, and the trigeminovascular pain pathway, with CGRP release, whose peptide levels are the highest, is activated during a migraine attack [14,15]. Monoclonal antibodies against CGRP (eptinezumab, fremanezumab and galcanezumab) or the CGRP receptor (erenumab) effectively prevent migraine attacks [16], which is at least comparable if not superior, to previous preventive drugs, with an unprecedented efficacy profile on adverse effects [17]. Real world Spanish data show that these drugs are equally effective in patients with medication overuse as in those without, and facilitate medication cessation [18].

Currently, more long-term follow-up is available, and evidence seems to point to related adverse events related being rare and with low rate of immunogenicity [19]. Constipation is a significant side effect of the use of monoclonal antibodies against CGRP in relation to Glucagon-Like Peptide-1 regulation [20]. Monoclonal antibodies against CGRP are, indeed, of added value for migraine prevention [17].

Several systematic reviews and meta-analyses have assessed the effects of monoclonal antibodies against CGRP versus no intervention or placebo [21,22,23,24,25]. However, there is no systematic review or meta-analysis with real-world data. In addition, systematic reviews and network meta-analyses have recently been conducted on randomized clinical trials (RCTs) with monthly migraine days (MMD) [26,27,28] and monthly headache days (MHD) [29] as outcomes; however, more head-to-head analyses are needed. Moreover, there are no studies comparing different effects of headache impact test (HIT-6) or monthly days with triptan use (TriD). Real-world studies complement clinical trials by generalizing the findings from clinical trial to general population [30]. Real-world studies may raise flags and/or new data on monoclonal antibodies against CGRP that were rarely observed or not described in RCTs. The results of RCTs cannot be generalized to the general population due to strict eligibility criteria. By performing a real-world metanalysis, we can analyze the effect of monoclonal antibodies against CGRP in a larger population and therefore draw more robust conclusions. Furthermore, network metanalysis, in the context of a systematic review, is a meta-analysis in which multiple treatments are compared, using both direct comparisons of interventions within RCTs and indirect comparisons between trials based on a common comparator [31]. A rigorous network meta-analysis could lead to therapeutic decision making on monoclonal antibodies against CGRP based on efficacy and safety. Thus, the aim of this protocol for a multiple-treatment systematic review and meta-analysis is to synthesize all available RCT and real-world evidence on the effect and safety of CGRP monoclonal antibodies in patients with migraine to establish the differences between them in (1) MMD, (2) MHD, (3) HIT-6, (4) TriD and (5) adverse events.

## 2. Materials and Methods

### 2.1. Protocol Register

This protocol for multiple-treatment systematic reviews and meta-analysis was reported in accordance with the PRISMA-P (Preferred Reporting Items for Systematic review and Meta-Analysis Protocols) statement [32] (Appendix A) and for real-world meta-analysis and network meta-analysis will be reported using the Meta-analysis of Observational Studies in Epidemiology (MOOSE) [33] and PRISMA-NMA [34,35] statements respectively. Additionally, the recommendations of the Cochrane Collaboration Handbook will be followed [36]. This multiple-treatment systematic review and meta-analysis was registered through the International Prospective Register of Ongoing Systematic Reviews (PROSPERO) [37], registration number CRD42021266322.

### 2.2. Ethics

Ethics committee approval is not required for this protocol. Data are not individualized.

### 2.3. Review Question

Following the PICO (population, intervention, comparison and outcome) strategy, our review question is as follow:Population: adult migraine patients;Intervention: monoclonal antibody against CGRP;Comparison: monoclonal antibody against CGRP and/or placebo;Outcome: MMD, MHD, HIT-6, TriD and adverse events.

### 2.4. Inclusion Criteria

-Real-world studies (for the real-world meta-analysis) and RCTs (for the network meta-analysis);-Articles in English or Spanish;-Chronic and/or episodic migraine adult patients (18 years or older) who have received any migraine treatment with monoclonal antibodies for migraine (erenumab, galcanezumab, fremanezumab and eptinezumab);-Articles with any of the following outcomes: MMD, MHD, HIT-6, TriD and adverse events;-Any monoclonal antibody against CGRP and/or placebo will be accepted as comparisons.

### 2.5. Exclusion Criteria

-Articles without complete measurement (value and dispersion) at baseline and after exposure, not completed even after contacting the authors;-Articles with a follow up length of less than three months;-Patients without chronic or episodic migraine;-Animal studies, letters and comments, review articles and editorials.

### 2.6. Information Sources and Search Strategy

Systematic searches of the MEDLINE Embase, CENTRAL, ClinicalTrials.gov, the WHO International Clinical Trials Registry Platform (ICTRP), Web of Science and Scopus databases will be conducted from their inception, following the PICO (population, intervention, comparison and outcome) strategy, which included the following terms: (migraine OR “episodic migraine” OR “chronic migraine” OR “ preventive treatment of chronic migraine” OR “prevention of episodic migraine” OR “migraine prevention” OR “all headache”) AND (erenumab OR galcanezumab OR fremazenumab or eptinezumab OR “monoclonal antibodies” or “humanized monoclonal antibody” OR “calcitonin gene-related peptide” OR “CGRP”) AND (“monthly migraine days” OR “migraine days OR MMD OR MMDs OR “change in average monthly migraine-days” OR “migraine days monthly” OR “headache-free days” OR “headache-free” “migraine headache” OR “headache days” OR “headache impact” OR “disability” OR efficacy OR effectiveness OR safety OR tolerability OR “adverse events” OR MIDAS OR “migraine disability assessment” OR MSQ OR MSQoL OR “migraine-specific quality of life questionnaire” OR “HIT-6” OR “headache impact test-6”). Study titles and abstracts will be screened for inclusion/exclusion criteria by two reviewers independently. There will be no filter limitations to the search

### 2.7. Study Selection

Once the search has been performed to identify eligible studies according to the inclusion criteria, the title and abstract will be assessed independently by two reviewers. Subsequently, abstracts and full-text articles that do not meet the inclusion criteria will be excluded. The full-text of the identified studies will be examined. Finally, two reviewers will verify the reasons why studies were included or excluded and present their process in a flow chart according to the PRISMA Statement [38]. (Figure 1). Mendeley (Mendeley, Elsevier, London) will be used for de-duplicated search results. The systematic literature search will be complemented by a review of the reference lists of the articles that were considered suitable for the real-world meta-analysis and network meta-analysis, as well as the reference lists of the already identified systematic reviews on this topic. Two authors of this protocol will screen all included databases independently. A third reviewer will resolve cases of initial reviewer disagreement.

The following information on the included studies will be provided independently by two authors: (1) reference (first author and publication year); (2) country in which the study data were collected; (3) population characteristics (sample size and percentage of women, mean age, pathology (CM/EM), years of migraine duration, previous and/or concomitant onabotulinum toxin-A); (4) intervention characteristics (monoclonal antibody against CGRP, dose administered and frequency, length of treatment and of follow-up); and (5) baseline levels of outcomes (MMD, MHD, HIT-6, TriDs and adverse events) (Table 1). When necessary to obtain missing information from the studies, the corresponding author will be contacted.

### 2.8. Assessment of Reporting Biases

We will use the ROBINS-I tool for nonrandomized experimental and single arm pre–post studies. This tool evaluates the risk of bias based on seven domains: confounding, selection of the study participants, measurement of interventions, deviations from intended interventions, missing data, measurement of outcomes and bias in the selection of reported results [39].

Furthermore, we will use the Cochrane Collaboration tool for risk of bias assessment (RoB2) of RCTs [40]. This tool is the most recommended because it reduces the subjectivity of the assessment, compared with other scales or checklists, by recording aspects of the RCT methods on which each trial is based according to prespecified criteria covering the following six bias domains: selection bias, performance bias, detection bias, attrition bias, reporting bias and other bias. This protocol will be performed by two researchers independently. A third reviewer will resolve cases of initial reviewer disagreement.

### 2.9. Grading the Quality of Evidence

The Grading of Recommendations, Assessment, Development and Evaluation (GRADE) tool will be use in order to assess the evidence quality and provide recommendations [41]. This tool includes the following five distinct steps for each outcome: assign an a priori classification of “high” to RCTs and “low” to observational studies; “downgrade” or “upgrade” the initial rating based on: risk of bias, inconsistency, indirect evidence, imprecision, publication bias, large effect, dose–response relationship and all plausible biases that only reducing an apparent treatment effect; assign the final rating of the quality of evidence as “high”, “moderate”, “low” or “very low”; address other influencing factors that affect the recommendation strength of a course of action; make a “strong” or “weak” recommendation [42].

### 2.10. Statistical Analysis

Two strategies are used depending on the studies selected, one for real-world studies and one for RCT. The effects of each intervention will be analyzed using STATA 15 (StataCorp, College Station, TX, USA).

#### 2.10.1. Real-World Meta-Analysis

The DerSimonian and Laird method [43] will be used to calculate the pooled estimates of the mean change difference (MD) for the monoclonal antibodies effect on MMD, MHD, HIT-6 and safety. The significance value of the pooled mean change will be estimated based on the 95% CI. Forest plots will be used to graphically depict the pooled MD for each main outcome.

#### 2.10.2. Network Meta-Analysis

The estimated effect of monoclonal antibodies (erenumab, galcanezumab, fremanezumab and eptinezumab) versus placebo or each other will be calculated using MD for each main outcome (MMD, MHD, HIT-6 and safety) and additional outcome (TriD) (for each RCT and outcome, the number needed to treat (NNT) will be calculated using the reported MD). Forest plots will be used to graphically depict the pooled MD for each main outcome and for each main adverse event in each treatment comparison. In addition, pooled MD estimates will be included along with their confidence intervals. Four-axis scatterplots will be generated displaying MD and NNT for each major outcome (x-axes) and RR (for safety) and NNT for serious adverse events (y-axes) by monoclonal antibodies against CGRP versus placebo [44]. Finally, to rank the most effective drug for each outcome and safety, the surface under the cumulative ranking (SUCRA) will be calculated and represented for each intervention. The SUCRA consists of assigning a numerical value between 0 and 1 to simplify the ranking of each intervention in the rankogram. The best intervention would score a SUCRA value close to 1, and the worst intervention would score a value close to 0 [45].

### 2.11. Heterogeneity Analysis

Statistical heterogeneity was examined by calculating the I^2^ statistic separately for each monoclonal antibody, which ranged from 0% to 100%. According to the I^2^ values, heterogeneity will be considered not important (0% to 30%), moderate (30% to 50%), substantial (50% to 75%), or considerable (75% to 100%) [46]. The corresponding *p* values were also considered. Finally, to determine the size and clinical relevance of heterogeneity, the Kendall’s τ statistic was calculated. A Kendall’s τ estimate of 0.04 may be interpreted as low, 0.14 as moderate and 0.40 as a substantial degree of clinical relevance of heterogeneity [47].

### 2.12. Publication Bias

Publication bias will be assessed both visually, by examining the funnel plots, and by performing Egger’s regression asymmetry test [48]. A level of 0.10 will be used to determine whether publication bias might be present.

### 2.13. Subgroup Analysis

Additionally, subgroup analyses (age, sex, risk of bias, medication overuse and previous and/or use of onabotulinum toxin-A) will be conducted. A random-effects meta-regression analysis will be performed to determine whether any variables modify the effect of the different monoclonal antibodies.

### 2.14. Sensitivity Analysis

Sensitivity analyses will be conducted to assess the robustness of the summary estimates and to detect whether any study accounts for a substantial proportion of the heterogeneity. For real-world meta-analysis, sensitivity analysis (systematic reanalysis by removing studies one at a time) will be conducted. The results of sensitivity analyses will be considered significant when the resulting estimates are modified beyond the 95% CI of the original pooled estimate.

## 3. Results

The results of the analysis will be analyzed according to country, female/male proportion, medication overuse, chronic migraine and/or episodic migraine and age. The results will be published as peer-reviewed articles.

The outcomes (mean differences, 95% CI) will be reported in the full reviews in narrative and tabular form. The outcomes to be reported are: MMD, MHD, HIT-6, TriDs and adverse events.

## 4. Discussion

The use of new monoclonal antibodies in clinical practice is a fact. Several studies have been conducted in comparison with placebo, showing the efficacy and safety of these drugs. However, it is a field that still needs to be investigated to establish whether there is a difference in efficacy and safety parameters between the four monoclonal antibodies. Patients and healthcare systems could benefit from this, with a better understanding of the best approach to manage of individual patients and better allocate the use of limited financial resources.

### Study Limitations

Meta-analysis may have potential limitations similar to those common to systematic reviews, i.e., publication and reporting bias. In addition, the real-world studies have no control group, so the blinded domain is classified as weak, and the outcome measure is based on a migraine diary completed by patients according to their subjective perceptions of their migraine episodes.

## 5. Conclusions

These two studies will provide information on the safety and efficacy of monoclonal antibodies indicated for migraine prevention. This information could be a particularly valuable tool for physicians to prescribe the appropriate monoclonal antibody for each patient according to evidence-based research and for healthcare systems to determine treatment and funding strategies and guidelines.

## Figures and Tables

**Figure 1 ijerph-19-01753-f001:**
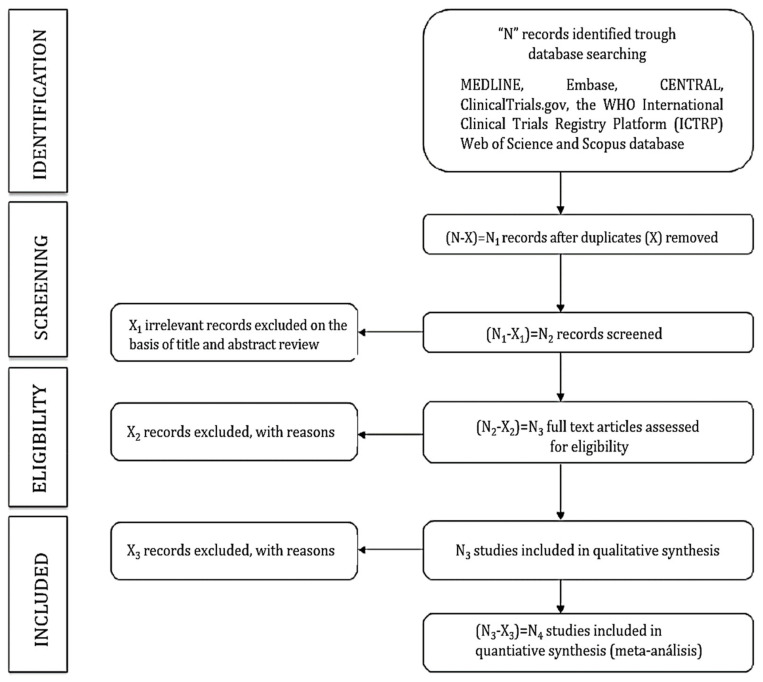
Flowchart of identification, screening, eligibility and inclusion of studies.

**Table 1 ijerph-19-01753-t001:** Characteristics of studies included.

				Population Characteristics				
Reference	Country	Sample Size	Mean Age	Pathology and Duration			Previous and/or Concomitant OnabotulinumtoxinA	
First author and year of publication	Country in which the study data were collected	Number of participants and percentage of women	Age (years) of the participants range or mean ± SD	Episodic Migraine or Chronic Migraine Years of migraine duration			Previous or actual administration of onabotulinumtoxinA	
Intervention characteristics				Outcome				
Monoclonal antibody against CGRP	Dose	Length		MMD	MHD	HIT-6	Tri-D	Adverse Effects
erenumab, galcanezumab, fremanezumab and eptinezumab	Dose administered and frequency	Length (months) of treatment and follow-up		Baseline and post exposure mean ± SD	Baseline and post exposure mean ± SD	Baseline and post exposure mean ± SD	Baseline and post exposure mean ± SD	Presence of any type of adverse effect and/or their grade.

## Data Availability

Not applicable.

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
