# Peer review of "The Safety and Efficacy of Calcitonin Gene-Related Peptide (CGRP) Monoclonal Antibodies for the Preventive Treatment of Migraine: A Protocol for Multiple-Treatment Systematic Review and Meta-Analysis"

_ijerph, 2022, doi:10.3390/ijerph19031753_

Round 1
Reviewer 1 Report
Dear Authors,
thank you for the opportunity to review this very valued work that provides a pre-published protocol, as is the standard in systematic review development. First of all, let me just briefly state that I enjoyed reading the protocol very much, and as a guideline and evidence synthesis methodologist, I appreciate that you took the time and effort to both register and publish a protocol before initiating work on the review itself. Please specify in your response when you are planning to start work on the review.
The protocol follows the highest current standard of reporting. The methods to be used in the review are presented clearly and the protocol has an excellent standard of language and writing.
I have a few minor suggestions to take the protocol just one small step further. This is only because to pre-specify all the steps of the review is an important part of the work. The suggestions I would like to emphasize is the inclusion of the assessment of the certainty in the evidence (GRADE Summary of Findings table), and revising the search sources. The rest are minor suggestions to improve the methods, reporting, and clarity.
Abstract: Delete numbering 1-4
Introduction: include a short definition of real-world studies and why you think it is important to add a meta-analysis of those to the existing ones of RCTs.
In general, based on the suggestions that follow, it would be very helpful to elaborate on what you state in the last section of the introduction (lines 64-75) – in my opinion, this information is essential for most readers (why you are doing this research, why these methods and what you think will be added to the pool of already existing systematic reviews).
Methods:
Include a review question using the PICO format (or another adequate format)
Elaborate on the inclusion and exclusion criteria (e.g., population – the type of migraine, length of follow-up, what comparisons will you accept, what outcomes will you be looking for). How will you handle any cointerventions (co-medication or non-pharmacological interventions for migraine) in RCTs and in the real-world studies?
Specify if there will be any limitations or filters to the search (e.g., date)
I understand you may not have access to all the databases, but as a minimum, and after consulting with our search specialist, I would suggest searching the following, whatever is feasible: MEDLINE, Embase and CENTRAL, ClinicalTrials.gov, the WHO International Clinical Trials Registry Platform (ICTRP). Web of Science, Scopus OK. Embase captures a lot of pharmacological studies not found elsewhere and CENTRAL is one of the top databases for RCTs. https://training.cochrane.org/handbook/current/chapter-04
Include the following information: Will you de-duplicate search results? Will you use any software for screening and extraction?
Will you search the reference lists of the included studies, and possibly also the reference lists of the already identified systematic reviews on the topic? If not, explain why.
What do you mean by this sentence and if it´s concerning the search, why is it in the study selection section? Line 123: „Two authors of this protocol will search all included databases independently.” I suspect you meant “screen” instead of “search”.
Data extraction: I suggest also extracting the length of the follow-up (if different from the length of treatment). Specify what length of follow-up will be used for analysis.
Why have you decided to use Quality Assessment Tool for Quantitative Studies for nonrandom-159 ized experimental and single-arm prepost studies? I suggest that you use the ROBINS-I tool for risk of bias assessment of the observational studies, especially since you are following the Cochrane Handbook methods. Or add an explanation why ROBINS cannot be used (I am not completely clear on what you mean by real-world studies, this term is used in various ways, and it does not necessarily mean that there is no control group as you state in the conclusion. In the most general sense, there is no controlled experiment but there may be comparisons used.)
How will you handle studies with a high risk of bias?
Include the Summary of findings – the GRADE assessment: https://training.cochrane.org/handbook/current/chapter-14 And state which outcomes you will report in the summary of findings table. Also, in the supplementary material the item 17 Confidence in cumulative evidence – this information is not provided in your protocol, so please add it in the protocol.
Correct the symbols used for heterogeneity: Change I2 to I2 (upper index), lines 196,197; also τ 2 (maybe Kendall’s τ reads better).
Include information on all the software you will use (screening, extraction, meta-analysis, publication bias statistics, etc.)
Results
Include information on how you will report the results – what will be reported in the full reviews, and in what form (tabular, narrative, which outcomes you will report, etc.).
Thank you very much for the time you will take to address these comments and I am looking forward to another read.
Best wishes
Reviewer 2 Report
The main topic of this work appears to be quite relevant at present and in the future management of migraine. The theoretical background is sufficiently contextualized in the introduction, but Authors have to clarify the use of “genetic” in the statement on line 37 “..an acute migraine attack is triggered by a wide range of genetic and environmental factors..”. Description of methods is exhaustive and concise. To sum up, the methodological approach of this work is consistent and adequate for the aim of the study, to notice that a few minor English revisions are needed to preserve the meaning of some sentences.
Reviewer 3 Report
The aim of the protocol submitted for review was to provide an overview comparing the effects and safety profile of different monoclonal antibodies in migraine patients. I believe that the presented protocol will significantly contribute to the improvement of monitoring the effectiveness and safety of the use of monoclonal antibodies in migraine.
Comments: The list of References requires unification and adaptation to the IJERPH Instruction for Authors.
